# Assessment of Acceptability and Initial Effectiveness of a Unified Protocol Prevention Program to Train Emotional Regulation Skills in Female Nursing Professionals during the COVID-19 Pandemic

**DOI:** 10.3390/ijerph19095715

**Published:** 2022-05-07

**Authors:** Vanesa Ferreres-Galán, María Vicenta Navarro-Haro, Óscar Peris-Baquero, Silvia Guillén-Marín, Jordi de Luna-Hermoso, Jorge Osma

**Affiliations:** 1Mental Health Unit of the Hospital Comarcal of Vinaròs, 12500 Vinaròs, Spain; ferreres_van@gva.es (V.F.-G.); jordideluna@gmail.com (J.d.L.-H.); 2Health Research Institute of Aragón, 50009 Zaragoza, Spain; operis@unizar.es (Ó.P.-B.); osma@unizar.es (J.O.); 3Department of Psychology and Sociology, University of Zaragoza, 44003 Teruel, Spain; guillenmarinsilvia@gmail.com

**Keywords:** prevention, emotion regulation skills, unified protocol, nursing professionals, COVID-19 pandemic

## Abstract

Healthcare professionals, especially women, have shown increases in anxious-depressive symptoms as a consequence of the COVID-19 pandemic. The aim of this pilot study was to evaluate the acceptability and preliminary effectiveness of a Unified Protocol (UP) prevention program to provide emotional regulation skills to cope with stressful situations. The sample consisted of 27 nursing professionals (100% women; mean age: 45.67; SD = 7.71) working in a Spanish public hospital during COVID-19, who were randomized to an immediate treatment group (ITG, *n* = 13) or to a delayed treatment group (DTG, *n* = 14, which served as the waiting list control group and received the program 5 weeks after the ITG had received it). The program consisted of five-weekly, two-hour, UP-based group sessions. Variables related to emotional symptomatology, emotional regulation, personality, burnout, and perceived quality of life were evaluated at the following time points: pre- and post-intervention and at 1-, 3-, and 6-month follow-ups. Statistically significant between-group differences showed lower emotional exhaustion and personal accomplishment in favor of the ITG after the intervention. Regarding the effect over time for all participants who received the UP (*n* = 27), statistically significant reductions were observed in neuroticism, personal accomplishment, and subjective distress caused by traumatic events (−0.23 ≤ *d* ≤ −0.73). A statistically significant interaction “Time*Condition” was found in anxiety, with increases in the DTG. Participants showed high satisfaction with the UP. These findings show good acceptability and preliminary effectiveness of the UP to reduce the emotional impact of the pandemic in female nursing workers.

## 1. Introduction

The COVID-19 pandemic is and has been a great global health challenge. The populations at greatest risk of suffering mental health problems due to the pandemic consequences include healthcare workers and, more specifically, front-line workers [1]. Healthcare workers are and have been addressing very severe job stressors for many months. For example, they have worked for more hours than usual, faced higher work overload and are at great risk of being infected, have underwent very strict safety protocols, and have been required to be highly concentrated and vigilant [2,3]. In turn, these factors have often placed healthcare workers under great stress [4,5]. As a consequence, burnout and fatigue have augmented in this population during the COVID-19 pandemic [6].

In this regard, several systematic meta-analyses and reviews have reported that the first wave of the COVID-19 pandemic was associated with a higher percentage of mental health problems in healthcare personnel, with emotional disorders (EDs; depression, anxiety, and related disorders; ref. [7]) being the most prevalent disorders in this population [8,9]. In fact, a study conducted by Luo et al. [10] that included healthcare personnel from 17 countries around the world reported a prevalence of anxiety and depression in this population group of 33% and 28%, respectively. In Spain, several studies were carried out with healthcare professionals during the first wave of the pandemic [11,12,13]. Findings are consistent with those obtained internationally in which a high proportion of anxious and depressive symptomatology has been observed, as well as high levels of stress. For example, a study developed by Alonso et al. [14] that included a sample of 9146 healthcare professionals from 18 healthcare institutions reported that 1 in 7 Spanish healthcare workers presented a probable mental disorder during the first wave of COVID-19. Specifically, 28.1% met criteria for a major depressive disorder, 22.2–24% for anxiety disorders (Generalized Anxiety Disorder, Panic Disorder, or Post-Traumatic Stress Disorder), and 6.2% for a substance use disorder. In turn, the risk of having a current mental disorder was higher in healthcare professionals who had been treating patients with COVID-19 more frequently and in those who had been in quarantine or isolation. Another important result was that the healthcare workers with the most risk to develop a mental disorder were those of the female gender and nursing professionals [14]. These findings suggest that there are great mental health needs to be met among healthcare personnel, especially in women, who should be considered a high-risk group for mental disorders. Thus, it is necessary to develop programs for the prevention and treatment of different mental health problems for this population, with EDs being the most prevalent psychological problems, and paying special attention to females.

Although several advances have been made, traditional treatments for EDs have shown some limitations, such as a high prevalence of comorbidity among EDs, with high costs for Public Health Services (direct and indirect), the ineffectiveness of some of the specific treatment protocols for these disorders, the high relapse rates at the end of treatment, and the economic and resource costs involved in the training and implementation of each of these protocols in the mental health services [15]. Based on these findings, and in order to resolve these problems, Barlow et al. [16] developed a treatment for EDs that emphasizes what the EDs have in common, rather than their differences. This treatment, the Unified Protocol for the Transdiagnostic Treatment of Emotional Disorders (UP), emphasizes that emotional regulation (ER) deficits (as intense emotional responses and aversive reactivity to emotional experiences) and avoidance behaviors are common to people with EDs [17]. The UP focuses on the common aspects of the different EDs, resulting in numerous advantages such as allowing the approach of people with comorbidity [18] as well as reducing costs related to the training of mental health workers in evidence-based treatment programs [19]. Moreover, because the UP consists of a modular intervention, it is considered more flexible and adaptable to different psychological problems [20,21] and treatment formats such as group, individual, face-to-face, and online [22,23].

Regarding the efficacy of the UP, a meta-analysis by Sakiris and Berle [24] and a recent systematic review by Carlucci et al. [25] showed that the UP, both in individual and group formats, significantly decrease anxiety and depression symptoms after treatment, and the effect sizes are comparable to those resulting from disorder-specific interventions. In addition to the effectiveness in emotional symptoms, the UP enhances overall functioning and quality of life [26,27]. Furthermore, preliminary research shows promising results of the effectiveness of preventive applications of the UP for nonclinical populations, such as students [28,29,30]. With respect to other psychological programs in Spain during the COVID-19 crisis for healthcare workers, a study showed that the most common component of these interventions in 36 hospitals was emotional regulation, which was implemented by psychoeducational and cognitive-behavioral techniques in individual interventions. Group interventions mainly used psychoeducation and mindfulness. However, no program effectiveness results were published [31]. Systematic reviews of preliminary studies show that interventions focused on building resilience may decrease perceived stress and burnout [32,33] and that cognitive behavior therapy and mindfulness-based interventions may be effective to treat symptoms of posttraumatic stress disorder (PTSD) due to COVID-19 experiences [34].

Therefore, and given the increases in EDs and the levels of stress and burnout in healthcare personnel after the COVID-19 pandemic, the application of the UP could be a useful psychological program to prevent EDs in this population and enhance coping with stressful situations. To our knowledge, there are no RCTs published that have investigated the effectiveness of the UP for healthcare workers and specifically nursing professionals, facing COVID-19. Thus, the main goal of this study was to evaluate the feasibility, acceptability, and preliminary effectiveness of the application of a brief UP prevention program (comparing an immediate treatment group and a delayed treatment group) to provide emotion regulation skills to cope with stressful situations among nursing professionals working in a public hospital in Spain during the health crisis due to the COVID-19 pandemic.

Specifically, our main goal was to evaluate the effectiveness of the UP to improve the severity of stress, anxiety, and depression (the primary EDs-related variables) after the intervention and at one-, three-, and six-month follow-ups. As secondary variables, we aimed to decrease emotional alterations in the context of a health crisis, such as emotional personality dimensions (neuroticism), difficulties in emotion regulation, and the impact of stressful events and professional burnout, as well as to increase quality of life in Spanish nursing professionals. In addition, we evaluated the acceptability of the UP-prevention program by assessing the participants satisfaction with the program. The main hypotheses that we expected were: statistically significant improvements in favor of the immediate treatment group (which received the UP-prevention program first) will be obtained after the intervention in the primary and secondary measures, compared with the delayed intervention group (waiting list). Once both groups have received the preventative program, the improvements obtained after the application of the UP will be maintained at one-, three-, and six-month follow-ups. The study sample will report high satisfaction scores regarding the UP-prevention program received.

## 2. Materials and Methods

### 2.1. Participants

The sample of this pilot study consisted of 27 professionals working at the nursing department of the Hospital Comarcal de Vinaròs (Spain), all of them women, with a mean age of 45.67 years (*SD* = 7.71, range 26–62). The participants were randomized to an Immediate treatment group (ITG) (*n* = 13) or a Delayed treatment group (DTG) (*n* = 14). The flow chart of the participants can be found in Figure 1.

### 2.2. Instruments

#### 2.2.1. Primary Instruments

*Depression, Anxiety, and Stress Scales* (*DASS*; [35,36]). This evaluates symptoms of depression, anxiety, and stress over the last 7 days, through 21 items with a 4-point Likert response scale ranging from 0 (not applicable to me at all) to 3 (very applicable to me, or applicable most of the time). Cronbach alpha values obtained in the present sample were: 0.90 for Depression, 0.58 for Anxiety, and 0.80 for Stress.

#### 2.2.2. Secondary Instruments

*Difficulties in Emotion Regulation Scale* (*DERS*; [37,38]). DERS consists of 28 response items ranging from 1 (Rarely) to 5 (Usually) that assess the levels of emotion dysregulation. Higher values reflect larger difficulties in emotion regulation. In this sample, Cronbach’s alpha for the DERS was 0.92.

*NEO-Five-Factor Personality Inventory* (*NEO-FFI*; [39]). It assesses the personality dimensions of neuroticism, extraversion, openness to experience, agreeableness, and conscientiousness through 60 items. In this study, only the variables of extraversion and neuroticism were measured. Participants answered a 5-point Likert scale scoring from 0 (Strongly disagree) to 4 (Strongly agree). The internal consistency of the NEO-FFI dimensions ranges between 0.82 and 0.90. Cronbach’s alpha values in the present sample were 0.82 for Neuroticism and 0.71 for Extraversion.

*Maslach Burnout Inventory* (*MBI*; [40,41]). This measure evaluates through 22 items the dimensions of emotional exhaustion, depersonalization, and personal accomplishment. The participants answered using a 7-point Likert response scale ranging from 0 (Never) to 6 (Every day). Cronbach’s alpha for the MBI (emotional exhaustion, depersonalization, and personal accomplishment) in the present sample were 0.91, 0.68, and 0.61, respectively.

*Impact of Event Scale-Revised* (*IES-R*; [42,43]). This consists of 22 items assessing the subscales of intrusion and avoidance caused by traumatic events. It uses a 5-point Likert-type response scale ranging from 0 (Never) to 4 (Always). Cronbach’s alpha for the IES-R in the present sample was 0.94 for the intrusion subscale and 0.89 for the avoidance subscale.

*European Quality of Life* (*EuroQol*; [44,45]). This measure is composed of 5 items that assess the subscales of mobility, self-care, activities of daily living, pain/discomfort, and anxiety/depression through a 5-point Likert scale ranging from 0 (no problem) to 4 (serious problems or inability to do anything). It also includes a visual analog scale (VAS) that assesses the overall health status perceived at the present moment and ranges from 0 (worst imaginable health status) to 100 (best imaginable health status). Cronbach’s alpha in the present sample is 0.76. In the present study, only the VAS was used in the analysis.

### 2.3. Procedure

This pilot study was conducted at Hospital Comarcal de Vinaròs (Spain), between April 2021 (pre-training of both groups) and December 2021 (6-month follow-up of the DTG group). The prevention program can be considered as the hospital’s 2nd measure aimed at preventing the development of EDs in this professional sector. This program was implemented to offer a solution to the needs that were discovered through the first measure offered by the hospital during the pandemic to the healthcare professionals, an over-the-phone psychological assistance service. Based on hospital internal data, the main requests for the psychological assistance service were made by nursing and auxiliary nursing staff (74%) and mostly women (85%), all of whom were at the front line of intervention in the course of the pandemic. They were a heterogeneous age group, with no significant differences in age. In the first wave of the pandemic, data from the psychological service reported that the psychological impact of COVID-19 was mainly caused by working conditions (lack of protective equipment, workload, continuous reorganizations, lack of knowledge of the virus disease process, etc.) as well as factors related to infected patients (high mortality rates, contact with suffering and death, etc.). However, in the second wave, the psychological demands were represented by the fear of repetition of situations and physical and emotional exhaustion.

According to this information, the participants were part of the nursing and auxiliary nursing staff of the hospital. The nursing supervisor provided information about the study to all members of the unit and those interested in participating signed the confidentiality, informed consent, and personal data protection documents.

No exclusion criteria were established, and the program was offered to all professionals from the nursing department who wished to participate. Of the 304 workers that were part of the nursing department (including nurses, auxiliary nursing care technicians, and anatomic pathology technicians), a total of 27 workers (8.88%) finally participated in the preventive program. Given that this was a service open to all personnel of the nursing department, no minimum number of participants was established. The inclusion criteria were: (a) to be part of the nursing department staff, (b) to be in active service, (c) to be fluent in the language in which the program will be applied (in this case, Spanish and/or Catalan), and (d) to be able to attend all the evaluation and intervention sessions. Subsequently, the two hospital psychologists sent the pre-program evaluation protocol (with codes assigned to each participant) to the supervisor who distributed them to each participant. After filling them out, the psychologists sent to the supervisor the list of participants with the random assignment (carried out through the Randomizer software by an independent researcher) to each condition of the study, as well as the schedule of sessions for both groups.

A delayed treatment control group design study was carried out to determine the effect of the UP-based prevention program. For this purpose, participants were divided into two conditions: Immediate treatment group (ITG) and Delayed treatment group (DTG).

The Immediate treatment group (ITG), formed by 13 participants, received the program consisting of the UP for the transdiagnostic treatment of EDs adapted to a preventive and brief format, which consisted of five weekly sessions of two hours in duration each.

The delayed treatment group (DTG)/waiting list consisted of 14 participants and received the preventive treatment immediately after the end of the intervention in the ITG.

There were two simultaneous evaluations in both groups, which consisted of pre-treatment (T0) and one month later (T1, coinciding with the ITG post-treatment, while the DTG had not initiated the intervention). After these simultaneous evaluations, the DGT received the UP intervention and their post-treatment evaluation was carried out (T2) approximately five weeks after the one carried out by the ITG (this extra evaluation was only carried out in the DTG condition). Once both groups received the UP-preventive program and performed their respective post-treatment evaluations, follow-ups were carried out at 1, 3, and 6 months (T3, T4, and T5, respectively). As participants in the DTG condition received treatment 5 weeks later than participants in the ITG condition, their follow-up evaluations also took place 5 weeks later than the follow-up evaluations conducted in the ITG condition.

As for the preventive program based on the UP, an adaptation of 5 sessions was developed in which the following components were taught in each session: (1) “Analysis of emotional experiences”, where work was done on the function of all emotions and the emotional experiences analysis (ARC); (2) “Living the present to facilitate emotional management”, where emotional awareness and mindfulness were addressed; (3) “Management of worries, uncertainty and fears: What to do or not to do”, in which cognitive flexibility and emotional behaviors vs. alternatives were the focus; (4) “Self-care; how to maintain reinforcing and meaningful activities in pandemic”, where values were clarified and pleasant activities were programmed; and (5) “Communication skills”, in which participants received training on assertiveness in communication with co-workers and patients.

### 2.4. Data Analysis

First, the sociodemographic data were analyzed through descriptive statistical analyses. Next, analyses of variance were carried out through the ANOVA test for continuous variables, and chi square tests for categorical variables, with the objective of analyzing if there were differences in the study variables between the ITG and the DTG at baseline. Subsequently, a Student’s *t*-test for related samples was performed to evaluate pre-post-treatment differences. Finally, a difference of means ANOVA was performed to compare the scores between the post-treatment of the ITG and that at the same time point with the scores of the DTG (without having received the intervention).

Following this, linear mixed model analyses were carried out with all participants (both ITG and DTG) in order to analyze the evolution of the scores over time. For the DTG condition, the most recent data before the start of treatment (T1) were included as baseline scores, to equalize the number of evaluation moments in the analyses. Given that the treatment, and the 1-, 3-, and 6-month follow-up evaluations in the DTG were carried out approximately five weeks later than those of the ITG, the variables “Condition” and the interaction “Condition*Time” were included as control variables within the linear mixed model, allowing us to analyze possible differences in the scores depending on whether they were immediate or delayed treatment groups.

Finally, post hoc analyses were carried out for those variables in which a statistically significant interaction effect “Condition*Time” was found, which consisted of replicating the linear mixed model but differentiating between the immediate treatment group and the delayed treatment group to analyze whether there were different evolution trajectories in the scores between groups over time. For all statistical analyses, effect sizes were calculated using Cohen’s d statistic, whose estimates are usually interpreted as small (*d* ≈ 0.2), medium (*d* ≈ 0.5), or large (*d* ≈ 0.8). All statistical analyses were carried out using SPSS software [46], p-values below 0.05 were considered statistically significant results, and, given that this was a pilot study and following the recommendations of the literature, a minimum of 10 participants per condition was expected in order to achieve a minimum of 80% statistical power, significance level of 0.05, and medium effect sizes (0.3 ≤ *d* < 0.7, [47]). In this study, a total of 13 and 14 participants per arm were obtained, so the minimum recommended values of statistical power would have been achieved. Finally, the participants’ satisfaction with the preventive program received was analyzed.

## 3. Results

### 3.1. Sociodemographic Outcomes and Virus Exposure of Participants

The sociodemographic information can be found in Table 1. Most participants of the sample were nurses or auxiliary nursing care technicians. Thirty point eight per cent of the participants in the ITG (*n* = 4) reported a psychological disorder in the past, specifically: depression (*n* = 2) and mixed anxiety depressive disorder (*n* = 2). As for the DTG, 35.7% (*n* = 5) reported a psychological disorder in the past, specifically: depression (*n* = 3), work stress (*n* = 1), and post-traumatic stress disorder (*n* = 1).

Concerning the presence of a current psychological disorder, 15.4% of the participants in the ITG (*n* = 2) reported one, specifically: depression (*n* = 1) and mixed anxiety depressive disorder (*n* = 1). On the other hand, 21.4% of the DTG (*n* = 3) informed of diagnoses of anxiety (*n* = 2) and depression (*n* = 1).

Finally, with respect to exposure to COVID-19, 85.2% (*n* = 23) of the participants live with at least one other person (range 1–4) and 70.4% of the participants have had a high COVID-19 exposure in the workplace (range 8–10).

### 3.2. Immediate Treatment Group and Delayed Treatment Group Results

First, the results showed no statistically significant differences in the scores at pre-treatment (T0) between the ITG and DTG (*p* > 0.05). Similarly, the results of the chi-square test showed no statistically significant differences between groups in the sociodemographic data (*p* > 0.05).

In terms of pre-post-treatment differences (T0-T1), the Student’s *t*-test for related samples showed statistically significant reductions in the ITG after receiving the preventive program for the variables DASS_Stress (*t* = 2.32, *p* =0.039, *Cohen’s d* = −0.48), DASS_Depression (*t* = 2.59, *p* = 0.024, *Cohen’s d* = −0.40), and MBI_Personal accomplishment (*t* = 4.96, *p* = 0.036, *Cohen’s d* = −0.76). With respect to the DTG, no statistically significant differences were found in any of the variables (*p* > 0.05) when comparing T0 and T1 scores (coinciding with the post-treatment evaluation of the ITG, and note that the DTG had not yet received the preventive program).

When comparing scores of the ITG and DTG in T1, statistically significant differences were found in MBI_Emotional Exhaustion (*F* = 4.66, *p* =0.042) and MBI_Personal accomplishment (*F* = 4.96, *p* =0.036), with lower scores in ITG. The mean variable scores for each of the groups are shown in Table 2.

### 3.3. Results of the Brief up Preventive Program over Time for All Participants

Regarding the evolution of the variables over time, and considering all participants who had received the preventive UP (both in ITG and DTG conditions), the results of the linear mixed model can be seen in Table 3. A statistically significant effect of time was found, with reductions in the variables Neuroticism (*F* = 2.58, *p* = 0.043, *dof* = 84.78, pre-treatment to 6-month follow-up *Cohen’s d* = −0.23), MBI_Personal accomplishment (*F* = 3.95, *p* = 0.005, *dof* = 86.25, pre-treatment to 6-month follow-up *Cohen’s d* = −0.65), IESR_Intrusion (*F* = 4. 91, *p* = 0.001, *dof* = 86.28, pre-treatment to 6-month follow-up *Cohen’s d* = −0.69), and IESR_Avoidance (*F* = 4.81, *p* = 0.001, *dof* = 87.15, pre-treatment to 6-month follow-up *Cohen’s d* = −0.73). Additionally, a statistically significant interaction “Time*Condition” was found in the DASS_Anxiety (*F* = 3.16, *p* = 0.018, *dof* = 90.78).

Post hoc analyses carried out for the DASS_Anxiety (see Table 4) showed a different evolution trajectory between the groups, finding a statistically significant effect of time on the DTG condition, with an increase in anxiety (*F* = 3.51, *p* = 0.014, *dof* = 45.58, pre-treatment to 6-month follow-up *Cohen’s d* = 0.49).

### 3.4. Satisfaction Results of the Brief up Preventive Program

Participants showed high satisfaction scores with the UP-prevention program received, with a mean of 8.17 out of 10 (SD = 7.71, range = 6.23–9.15). Regarding the qualitative information collected, 59.26% (*n* = 16) of the participants expressed the need for a greater number of sessions and a longer duration of the program, and 44.44% (*n* = 12) showed their satisfaction with the program reporting: “*I found the program very useful, I would recommend it to anyone, I would sign up again*”, “*I really liked it, more things like this should be done*”, “*It should be offered periodically to health professionals, not only in pandemics*”.

## 4. Discussion

Research has shown that the prevalence of EDs has significantly increased in healthcare workers as a consequence of the COVID-19 pandemic [11,12,13,14]. The UP is one of the most effective treatments to address EDs by reducing emotional symptomatology and increasing quality of life in different clinical and nonclinical populations [24,25,26,27]. Recent studies have shown preliminary evidence of the UP as a preventive program of EDs in the general population [28,29]. However, to our knowledge, this is the first study to evaluate the UP in professionals of a nursing department exposure to COVID-19. Therefore, the general aim of this study was to investigate the acceptability and effectiveness of a brief five-week UP prevention program group in order to help Spanish nursing professionals cope with stressful situations during the COVID-19 pandemic.

We hypothesized to find statistically significant differences in favor of the ITG (the group who first received the UP-prevention program) compared to the DTG (waiting list) in reducing the severity of stress, anxiety, and depression, as well as other related variables such as the impact of stressful situations, burnout, difficulties in emotion regulation, and emotional personality dimensions. In addition to reducing psychopathology, we also expected that the ITG would be superior to the DTG at increasing quality of life in the Spanish nursing workers. Another hypothesis proposed was that once both groups had received the preventative program, the results obtained after the application of the UP will be maintained at one-, three-, and six-month follow-ups.

Regarding the results of our study, first, it is important to highlight that between 15.4% and 21.4% of the nursing workers interviewed for the study reported having at least one ED at baseline, and around 70% of the participants had experienced high COVID-19 exposure at work. Another important issue is that, although the study was open to any nursing worker of the hospital, 100% of the study sample were women. These findings correspond with data of a previous study with similar populations [11,12,13,14] and suggest the need to develop programs to reduce EDs in this vulnerable population.

With respect to the findings in the ITG and DTG before and after the treatment (before the DTG had received the program), reductions in anxiety, depression, as well as personal accomplishment were found in ITG but not in DTG. Similar outcomes of other UP preventive programs in reducing depression and anxiety symptoms for adolescents and college students [27,30] have been found. Moreover, although there is little published literature on the effectiveness of psychological interventions for this specific population, several quasi-experimental studies show that programs focused on coping with stress and improving resilience in healthcare professionals during the COVID-19 crisis encountered pre-post improvements in perceived stress and burnout and were identified as potentially suitable and useful for improving psychological functioning [32,33]. However, although improvements were found in depression and anxiety in our study, a possible explanation for the decrease in the facet of burnout of personal accomplishment (i.e., feelings of competence and successful achievement in our work [40,41]) after treatment may have to do with a peak increase in COVID-19 in one of the waves, where the pressure of assistance was highest and personal accomplishment could have been difficult to maintain.

Regarding the evolution of the study variables over time, and considering all participants who had received the preventive UP (both in ITG and DTG conditions), a statistically significant effect of time was found to result in reductions in neuroticism and personal accomplishment with medium to large effect sizes. In addition, a significant improvement was found in subjective distress caused by traumatic events (e.g., COVID-19), as measured by intrusion (i.e., nightmares, visual images of the trauma, intrusive thoughts about the traumatic event) and avoidance (i.e., deliberate efforts to not talk about the event, not think about the event, and to avoid reminders of the event) subscales of the IES-R [17,18]. This last finding is interesting as symptoms of post-traumatic stress disorder have been identified as a common symptom directly caused by COVID-19 exposure in healthcare professionals, and the urgent need to develop programs to address this problem has been suggested [48]. To our knowledge, there are no studies that have investigated the effectiveness of the UP to decrease symptoms of post-traumatic stress during COVID-19, and these results show that only by applying a brief UP preventive program was a significant reduction in these severe and disabling symptoms achieved. Regarding other psychological interventions to reduce PTSD symptoms due to COVID-19 experiences, a systematic review found that the most feasible and effective treatment program for healthcare professionals with PTSD is still unclear; however, cognitive behavior therapy and mindfulness-based interventions have shown the most significant effects based on current limited evidence [34].

In addition to these results, post hoc analyses showed a different evolution trajectory between ITG and DTG, finding a statistically significant effect of time on the DTG condition in increasing anxiety over time. If we analyze the results by time points, we can observe that this significant difference becomes much worse at T5, coinciding with the Christmas COVID-19 wave peak. A future research direction would be to conduct longitudinal studies and consider contextual factors (such as peaks of COVID-19) that may influence outcomes of preventive interventions.

On the other hand, contrary to our hypothesis, we did not find significant between-group differences in quality of life and emotional regulation over time. Although there was a tendency to increase after treatment, the improvements were not maintained long-term. This is surprising as previous studies of the UP in other populations have shown improvements in both variables over time [24,26,27]. Perhaps this might be due, as suggested by participants, to the brief intervention received and/or to the contents we chose to be included in the program. In this sense, we included the contents of the original UP module numbers 2, 3, 4, and 5 and added two new skills: pleasant activities and assertiveness training, both relevant to enhance stress coping during the pandemic situation. We did not include UP modules 1 (setting goals and maintaining motivation), 6 (understanding and confronting physical sensations), 7 (emotion exposures), and 8 (recognizing accomplishments and looking to the future), because of the nonclinical nature of the participants. It is possible that the other modules and components might have different results in quality of life and emotional regulation variables. Future studies may focus on developing different programs or replicating the one described in the present study to test their effectiveness at improving these specific variables.

A final hypothesis is that the study sample will report high acceptability of the UP-prevention program received. Findings showed high satisfaction scores with the UP-prevention program received, nonetheless, participants suggested the need to increase the number of sessions and the duration of the program. Another interesting outcome is that no drop-outs happened during the intervention. This was the first evidence-based preventive program conducted at the hospital for workers of the nursing department; therefore, the results of acceptability are promising given the brevity of the program.

Despite these promising findings in reducing emotional symptoms, this study has some limitations. First, all participants were women. This could be explained by the fact that most nurses (84.1% of certificated nurses in Spain in 2020) are women [49]. However, it may be important to replicate these results in men. Secondly, the size of the sample was small. Future studies should conduct RCTs with a bigger sample. In addition to this, some of the measures used in this study have shown Cronbach’s alpha values below the recommended values (values below 0.70, [50]), specifically in the anxiety subscale of the DASS (Cronbach’s alpha 0.58) and the Depersonalization and Personal accomplishment subscales of the MBI (Cronbach’s alpha 0.68 and 0.61, respectively). These sub-dimensions were the ones that presented the lowest Cronbach’s alpha indices in the validations (i.e., in the Spanish validation of this instrument, the Anxiety sub-dimension also presented the lowest value with a Cronbach’s alpha of 0.70 [36], or the Personal accomplishment in the original validation of the MBI showed a Cronbach’s alpha of 0.71 [40]). In addition, the number of observations in this study (*n* = 27) perhaps has a decisive influence on these low scores, so the results of these instruments should be considered with caution. Finally, given that there were changes in the pressure of assistance during the different waves of COVID-19, a limitation of this study was not to control for this contextual variable. As previously suggested, this may be a future research direction.

## 5. Conclusions

The results of this study show statistically significant reductions over time in neuroticism and subjective distress caused by traumatic events for all the female workers of the nursing department who received the UP-prevention program with medium to large effect sizes. A statistically significant interaction of time by condition was found in symptoms of anxiety, with greater anxiety in the DTG, which may be due to the changes in care pressure during the different waves of COVID-19. In addition, 100% of participants finished the program and showed high satisfaction with the UP received, highlighting the need to increase the number of sessions and the duration of the program. In conclusion, these findings suggest good acceptability and preliminary effectiveness of the UP to improve emotional symptomatology in female nursing professionals. Studying contextual factors such as the increases in pressure of assistance during COVID-19 waves for future longitudinal studies may be useful to determine the impact of the prevention programs.

## Figures and Tables

**Figure 1 ijerph-19-05715-f001:**
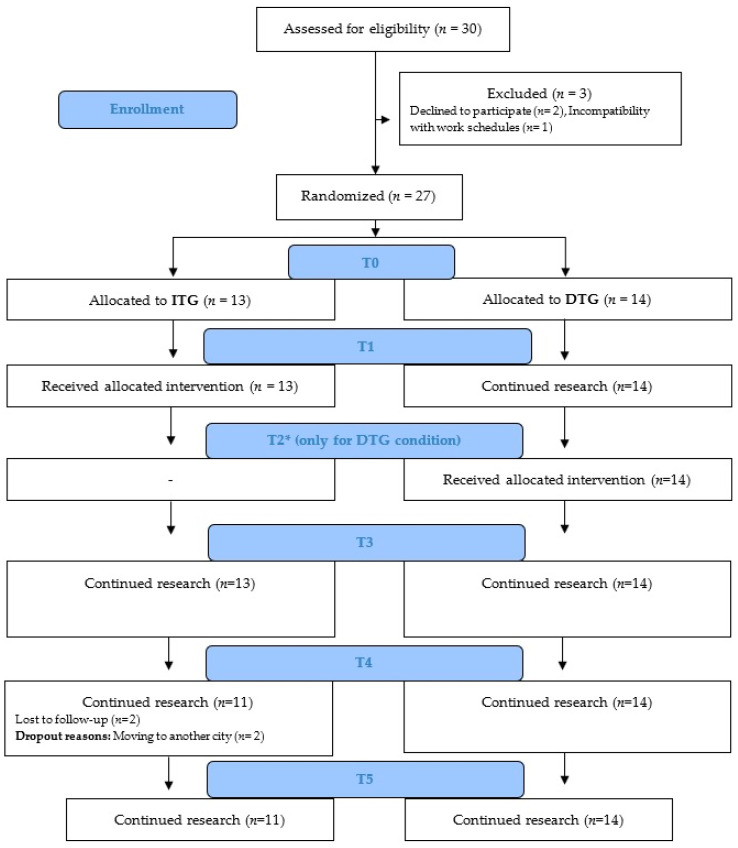
CONSORT diagram illustrating participant flow in the study. Note: ITG: Immediate treatment group; DTG: Delayed treatment group; T0: Pre-treatment; T1: Immediate treatment group post-treatment; T2: Delayed treatment group post-treatment; T3: 1-month follow-up; T4: 3-month follow-up; T5: 6-month follow-up.

**Table 1 ijerph-19-05715-t001:** Baseline socio-demographic characteristics of the sample across treatment conditions (N = 27).

	ITG (*n* = 13) *n* (%)	DTG (*n* = 14) *n* (%)	TOTAL (N = 27) *n* (%)
Marital Status			
Married/living with partner	10 (76.9)	11 (78.6)	21 (77.8)
Single	3 (23.1)	2 (14.3)	5 (18.5)
Widowed	0 (0.0)	1 (7.1)	1 (3.7)
Occupation			
Nurse	6 (46.2)	8 (57.1)	14 (51.9)
Auxiliary nursing care technician	7 (53.8)	5 (35.7)	12 (44.4)
Anatomic pathology technician	-	1 (7.1)	1 (3.7)
Hospital Unit			
Internal Medicine B	4 (30.8)	4 (28.6)	8 (29.6)
Internal Medicine A	3 (23.1)	3 (21.4)	6 (22.2)
COVID-19 Surgery	3 (23.1)	0 (0.0)	3 (11.1)
Emergencies	0 (0.0)	2 (14.3)	2 (7.4)
COVID-19 plant	1 (7.7)	0 (0.0)	1 (3.7)
Others	2 (15.4)	5 (35.6)	7 (25.9)
Work experience			
More than 5 years	11 (84.6)	13 (92.9)	24 (88.9)
2–5 years	2 (15.4)	1 (7.1)	3 (11.1)
Virus Exposure (0 “no contact” to 10 “close contact”)			
Less than 8	4 (30.8)	4 (28.6)	8 (29.6)
More than 8	9 (69.2)	10 (71.4)	19 (70.4)
8	3 (23.1)	0 (7.1)	3 (11.1)
9	1 (7.7)	1 (7.1)	2 (7.4)
10	5 (38.5)	9 (64.3)	14 (51.9)
COVID infection			
No	10 (76.9)	12 (85.7)	22 (81.5)
Yes	3 (23.1)	2 (14.3)	5 (18.5)
Severity of Symptoms (0 “no symptoms” to 10 “severe symptoms”)			
0	1 (33.3)	1 (50.0)	2 (40.0)
2	1 (33.3)	1 (50.0)	2 (40.0)
3	1 (33.3)	0 (0.0)	1 (20.0)
Family members infected			
No	8 (61.5)	10 (71.4)	18 (66.7)
Yes	5 (38.5)	4 (28.6)	9 (33.3)

Note: ITG: Immediate treatment group; DTG: Delayed treatment group; Internal Medicine (A, B): group of medical specialty focused on the global treatment of diseases.

**Table 2 ijerph-19-05715-t002:** Means and standard deviations of the variables over time in the immediate and delayed treatment groups. (N = 27).

			T0	T1	T2	T3	T4	T5
			M (SD)	M (SD)	M (SD)	M (SD)	M (SD)	M (SD)
Primary outcomes	DASS_Stress	ITG	6.62 (4.52)	4.54 (3.99)	-	4.77 (4.04)	5.18 (4.79)	4.18 (3.19)
		DTG	7.36 (3.59)	5.42 (2.23)	5.29 (2.09)	5.45 (2.66)	5.17 (2.44)	7.17 (3.19)
	DASS_Anxiety	ITG	4.00 (3.56)	2.15 (1.95)	-	1.85 (2.34)	2.27 (3.90)	1.36 (1.96)
		DTG	3.50 (1.7)	2.92 (2.15)	3.00 (2.83)	2.18 (1.72)	1.83 (2.21)	4.17 (2.86)
	DASS_Depression	ITG	5.85 (5.8)	3.69 (5.02)	-	3.08 (3.57)	4.09 (4.61)	4.27 (4.34)
		DTG	5.00 (4.76)	2.67 (1.92)	4.14 (3.7)	2.70 (2.87)	3.08 (2.31)	3.67 (2.77)
Secondary outcomes	DERS	ITG	58.92 (17.54)	54.23 (18.21)	-	51.54 (15.69)	50.55 (14.47)	49.36 (13.84)
		DTG	56.57 (14.85)	55.75 (11.14)	55.23 (12.06)	49.4 (9.99)	55.27 (15.51)	53.83 (12.08)
	Neuroticism	ITG	21.77 (9.64)	21.46 (8.25)	-	18.17 (8.70)	18.00 (9.19)	18.00 (9.30)
		DTG	21.79 (4.64)	19.00 (5.77)	19.36 (5.53)	19.5 (5.10)	19.5 (5.65)	19.25 (4.96)
	Extraversion	ITG	25.46 (5.09)	26.92 (5.92)	-	28.42 (6.99)	28.9 (5.57)	28.00 (6.45)
		DTG	24.21 (6.96)	25.92 (4.89)	26.00 (6.8)	27.67 (7.78)	26.67 (7.02)	27.08 (6.52)
	MBI_Emotional Exhaustion	ITG	11.00 (9.21)	9.69 (6.97)	-	10.15 (7.36)	11.36 (5.73)	13.45 (9.70)
		DTG	16.00 (11.64)	17.17 (10.17)	16.21 (7.68)	15.3 (8.65)	14.18 (8.81)	17.33 (11.11)
	MBI_Depersonalization	ITG	5.85 (4.24)	5.69 (3.84)	-	5.08 (4.09)	5.45 (5.35)	5.55 (4.18)
		DTG	4.79 (5.18)	5.33 (3.87)	4.57 (3.80)	2.91 (2.39)	3.42 (3.32)	3.92 (4.78)
	MBI_Personal accomplishment	ITG	40.92 (4.92)	35.15 (6.57)	-	38.23 (6.37)	40.00 (3.9)	36.82 (5.51)
		DTG	38.79 (5.92)	40.42 (5.07)	38.57 (7.90)	40.6 (4.43)	39.18 (6.6)	37.42 (6.69)
	IESR_Intrusion	ITG	23.15 (13.63)	19.85 (13.56)	-	15.33 (13.25)	16.73 (12.64)	14.27 (11.88)
		DTG	18.5 (9.59)	20.33 (9.16)	19.85 (8.22)	18.45 (13.02)	15.42 (8.51)	14.5 (6.97)
	IESR_Avoidance	ITG	15.08 (7.58)	13.08 (6.60)	-	11 (6.78)	11.36 (6.87)	9.82 (7.21)
		DTG	11.21 (5.94)	13.75 (6.31)	12.79 (4.12)	10.18 (6.21)	11.58 (5.43)	9.75 (4.65)
	EuroQol	ITG	74.23 (18.13)	76.54 (17.37)	-	82.67 (13.61)	79.55 (15.92)	81.36 (16.45)
		DTG	77.86 (7.77)	78.00 (7.51)	83.75 (7.72)	86.36 (5.52)	78.17 (24.48)	78.33 (14.51)

Note: ITG: Immediate treatment group; DTG: Delayed treatment group; T0: Pre-treatment; T1: Immediate treatment group post-treatment; T2: Delayed treatment group post-treatment; T3: 1-month follow-up; T4: 3-month follow-up; T5: 6-month follow-up; DASS_Stress, DASS_Anxiety, and DASS_Depression: Depression, Anxiety, and Stress Scales dimensions; DERS: Difficulties in Emotion Regulation Scale; Neuroticism and Extraversion: NEO-Five-Factor Personality Inventory dimensions; MBI_Emotional Exhaustion, MBI_Depersonalization, and MBI_Personal accomplishment: Maslach Burnout Inventory dimensions; IESR_Intrusion and IESR_Avoidance: Impact of Event Scale-Revised; and EuroQol: European Quality of Life.

**Table 3 ijerph-19-05715-t003:** Main effects of the linear mixed models.

	Time			Condition			Time*Condition		
	*F*	*p*	*Cohen’s d*	*F*	*p*	*Cohen’s d*	*F*	*p*	*Cohen’s d*
DASS_Stress	1.04	0.390	0.41	0.02	0.895	0.18	1.59	0.184	0.50
DASS_Anxiety	1.94	0.110	0.56	0.38	0.541	0.25	**3.16**	**0.018**	0.71
DASS_Depression	1.26	0.290	0.45	0.99	0.328	0.40	1.94	0.110	0.56
DERS	2.36	0.060	0.61	0.01	0.917	0.04	0.97	0.425	0.39
Neuroticism	**2.58**	**0.043**	0.64	0.31	0.583	0.22	1.61	0.179	0.51
Extraversion	2.09	0.089	0.58	0.00	0.992	0.00	0.33	0.859	0.23
MBI_Emotional Exhaustion	1.19	0.321	0.44	3.03	0.093	0.70	0.62	0.653	0.31
MBI_Depersonalization	0.80	0.527	0.36	1.70	0.204	0.52	0.20	0.939	0.18
MBI_Personal accomplishment	**3.95**	**0.005**	0.80	0.52	0.476	0.29	1.14	0.343	0.43
IESR_Intrusion	**4.91**	**0.001**	0.89	0.29	0.597	0.22	1.26	0.290	0.45
IESR_Avoidance	**4.81**	**0.001**	0.88	0.39	0.535	0.25	0.10	0.983	0.13
EuroQol	1.88	0.121	0.55	0.43	0.520	0.26	0.64	0.638	0.32

Note: OASIS: DASS_Stress, DASS_Anxiety, and DASS_Depression: Depression, Anxiety, and Stress Scales dimensions; DERS: Difficulties in Emotion Regulation Scale; Neuroticism and Extraversion: NEO-Five-Factor Personality Inventory dimensions; MBI_Emotional Exhaustion, MBI_Depersonalization, and MBI_Personal accomplishment: Maslach Burnout Inventory dimensions; IESR_Intrusion and IESR_Avoidance: Impact of Event Scale-Revised; and EuroQol: European Quality of Life; *p*-values < 0.05 are shown in bold.

**Table 4 ijerph-19-05715-t004:** Post hoc analyses for the Time*Condition.

			Main Effects							*Cohen’s d*		
		T0	T1	T2	T3	T4	T5	*F*	*p*	Pre-T-to-Post-T	Post-T-to-6-MFU	Pre-T-to-6-MFU
		M (SD)	M (SD)	M (SD)	M (SD)	M (SD)	M (SD)					
DASS_Anxiety	ITG	4.00 (3.56)	2.15 (1.95)	-	1.85 (2.34)	2.27 (3.90)	1.36 (1.96)	2.20	0.084	−0.64	−0.40	−0.92
	DTG	3.50 (1.7)	2.92 (2.15)	3.00 (2.83)	2.18 (1.72)	1.83 (2.21)	4.17 (2.85)	**3.51**	**0.014**	0.03	0.41	0.49

Note: ITG: Immediate treatment group; DTG: Delayed treatment group; T0: Pre-treatment; T1: Immediate treatment group post-treatment; T2: Delayed treatment group post-treatment; T3: 1-month follow-up; T4: 3-month follow-up; T5: 6-month follow-up; DASS_Anxiety: Depression, Anxiety, and Stress Scales; *p*-values < 0.05 are shown in bold.

## Data Availability

Not applicable.

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
