# Peer review of "Assessment of Acceptability and Initial Effectiveness of a Unified Protocol Prevention Program to Train Emotional Regulation Skills in Female Nursing Professionals during the COVID-19 Pandemic"

_ijerph, 2022, doi:10.3390/ijerph19095715_

Round 1

Reviewer 1 Report

This is an interesting manuscript that I enjoy reading, the thematic is original especially because it is evaluated the effectiveness of a psychological intervention programme during COVID-19 pandemic as a way to support e promote the wellbeing of healthcare personnel. I think the manuscript need some revisions before it can be accepted for publication

I have made some recommendations/comments below:

Abstract:

I suggest to introduce the measures evaluated as in the results several dimensions appeared and it should be reported to which test they referred to.

Measures

I noted that several chronbach’s alpha are lower than .70 that is generally used as minimum acceptable value, this represent a strong limit that should be commented by Authors and inserted in the limit section.

Participants

In the inclusion criteria the female gender was not reported so I can suppose that the participation of only female workers was casual. Is it correct? There were male workers in the department selected for the participant enrolment? The Authors reported that female healthcare workers are more exposed to negative consequences of pandemic forthermore in the title is highlighted that is an evaluation on only female participants and this remark in the title can let the reader think that the Author choose to invite to participate only female. I think this point should be better explained.

Furthermore, as regards the sample the Authors did not report how they determined the number of participants needed to test the study hypothesis (effect size).

Figure 2

In ITG group T2 is lacking and it passes from 1 to 3

Discussion

I think the discussion should be deepened better exploring the comparison of present findings with previous existing results.

Limits

The sample size is a very strong limit of the present study, I think the author should be careful in their conclusion and I suggest to insert in the title the term pilot study.

Reviewer 2 Report

The aim of the study was to evaluate the acceptability and preliminary effectiveness of a Unified Protocol prevention program to provide emotional regulation skills to cope with stressful situations in group 27 female health workers.

I would suggest modifying the title of the manuscript, as the current wording does not fully reflect the essence of the study. I propose the following wording for the title "Assessment of acceptability and initial effectiveness Unified Protocol prevention program to train emotional regulation skills in female health care workers during the COVID-19 pandemic". In the wording that I propose, the title is longer, but in my opinion it reflects the purpose of the study.

Article is well-organized. The layout of the article is typical of original papers and includes all required parts, but some sections require improvement.

In the "Abstract" section, there is no information on the psychological measures used named in the manuscript in section 2.2 "Secondary instruments". In the same section ("Abstract") it is necessary to explain the terms "immediate treatment group (ITG)” and “a delayed treatment group (DTG)” (lines 16-17).

In the section 2.3 “Procedure the statement” the following sentence fragment: "No exclusion criteria were established" (line 193) requires justification.

In my opinion, the study scheme, groups description, and terms related to the interventions used presented in section 2.3 “Procedure the statement” and presented in Figures 1 and 2, require a more precise description. In their current form, there are discrepancies between the description and the figures and raise doubts in interpretation. In addition, both figures (1 and 2) show a similar range, but neither is clear enough.

Moreover, the symbols placed in the figures must be explained below them. This applies to the following markings: ITG, DTG, T0, T2, T3, T4, T5.

In the caption for figure 1, unnecessary spaces should be removed.

In the section 2.4 “Data analysis” the naming of variables should be standardized: “Condition” and “Experimental Condition”.

"Note" below tables 2 and 4 need to be corrected: "T3: Delayed treatment group post-treatment" (line 300) should be replaced with "T2: Delayed treatment group post-treatment". In addition, in "Note" below table 2 the double colon after “T1” (line 299) should be eliminated.

Section 2.4 “Data analysis” – for which p-value significant levels was considered?

Section 2 “Materials and Methods” it is worth supplementing with the date of the study.

The study was conducted in a small group, which the authors indicated as a limitation of the study. Therefore, it would be worth supplementing the text with the information what percentage of all people who could take part in the study was the study group - described in the manuscript.

Reference list has not been fully prepared according to the rules for the Journal and needs minor improvement, among others: unnecessary capital letters in the title of the articles should be eliminated, abbreviated journals name should be inserted, abbreviated journal names should start with capital letters. Furthermore, the name “pietro” in line 575 should start with a capital letter. In item 7 of the reference list (line 482), remove unnecessary spaces before commas.

Overall, the Authors achieved their goals and drew the correct conclusions. Therefore, the results obtained by the Authors and conclusions should be published.

Reviewer 3 Report

Dear authors, thank you for the opportunity to review this study. Undoubtedly, the issue raised in them is very important, I think not only for healthcare professionals. I have enormous concerns about a very small test group. What does the power of the test look like? Why was it decided to only involve women? What are the results if we analyzed people with no previous psychiatric conditions - is the answer the same? In addition, in the discussion I miss a comparison of your method with other available? Are there any? It is worth paying attention to this.

Round 2

Reviewer 3 Report

The authors fully commented on the previous suggestions. Despite the small size of the group, it meets the publication criteria. However, in the methodology section, please complete the information about the power of the test.
